# Stabilization of cultural innovations depends on population density: Testing an epidemiological model of cultural evolution against a global dataset of rock art sites and climate-based estimates of ancient population densities

**Richard Walker**[1‡]*, **Anders Eriksson**[2‡]*, **Camille Ruiz**[3], **Taylor Howard Newton**[1], **Francesco Casalegno**[1]

**1** Blue Brain Project, Ecole Polytechnique Fédérale de Lausanne, Lausanne, Switzerland, **2** Institute of Genomics (cGEM), University of Tartu, Tartu, Estonia, **3** Ateneo de Manila University, Manila, The Philippines

‡ RW and AE are joint authors.
* richard.walker@epfl.ch (RW); anders.eriksson@ut.ee (AE)

**Data Availability Statement:** Our rock art data set is included in the Supplementary Information, in

## Abstract

Demographic models of human cultural evolution have high explanatory potential but weak empirical support. Here we use a global dataset of rock art sites and climate and genetics-based estimates of ancient population densities to test a new model based on epidemiological principles. The model focuses on the process whereby a cultural innovation becomes endemic in a population, predicting that this cannot occur unless population density exceeds a critical threshold. Analysis of the data, using a Bayesian statistical framework, shows that the model has stronger empirical support than a proportional model, where detection is directly proportional to population density, or a null model, where rock art detection ratios and population density are independent. Results for different geographical areas and periods are compatible with the predictions of the model and confirm its superiority with respect to the null model. Re-analysis of the rock art data, using a second set of independent population estimates, again supports the superiority of the model. Although the available data is sparse and the analysis cannot exclude all possible sources of bias, this is evidence that population density above a critical threshold may be a necessary condition for the maintenance of rock art as a stable part of a population's cultural repertoire. Methods similar to those described can be used to test the model for other classes of archaeological artifact and to compare it against other models.

## Introduction

It is widely accepted that the complexity and diversity of human cultures are a result of Cumulative Cultural Evolution (CCE), enabled by humans' unique neuroanatomy and cognitive

both PDF and CSV formats. The population estimates used in our modeling are available at https://osf.io/dafr2/ (Eriksson data) and https://climatedata.ibs.re.kr/grav/data/human-dispersal-simulation (Timmermann data). To facilitate other users all data, analysis parameters and code used in the paper are available in a github repository (https://github.com/rwalker1501/cultural-epidemiology.git).

**Funding:** The authors received no specific funding for this work.

**Competing interests:** The authors have declared that no competing interests exist.

capabilities, especially their skills in "cultural learning" [1–3]. However, the old idea that modern human capabilities emerged suddenly as a result of an advantageous mutation some 50,000 years ago [4,5] is no longer accepted and many allegedly modern features of human behavior and cognition are now believed to be more ancient than previously suspected [6–9]. There is, furthermore, no evidence that variations in brain size, brain morphology or innate capabilities explain any aspect of the spatiotemporal patterning of human cultural evolution over the last 50,000 years. Against this background, demographic models of CCE [10] assign a determining role to variations in population size, density and structure. Such models suggest that larger, denser, better connected populations produce more numerous, and more complex innovations than smaller ones [11], are less vulnerable to stochastic loss of unique skills [12,13], and are more capable of exploiting beneficial transmission errors during social learning [14]. They also suggest that well-connected metapopulations produce faster cultural innovation than metapopulations with weaker connections among subpopulations [15].

Demographic models could potentially provide valuable explanations for spatiotemporal patterning in the global archaeological record and several authors have used them for this purpose e.g. [14,16]. However, the assumptions and conclusions of the models are hotly contested [17–19]. Empirical studies are sparse and inconclusive, with some supporting an important role for demography (e.g. [14,15,20,21]), while others find no evidence for such a role [22–26].

CCE involves the inception, diffusion and selective retention of cultural innovations (e.g., innovations in social practices, beliefs, and technologies). In the process, a subset of the innovations become *endemic*–providing a stable foundation for future innovation. This paper presents a model of this process, inspired by SIR (Susceptible-Infectious-Recovered) models in epidemiology [27–29]. We test the model for the case of parietal rock art.

## The model

The model used in this study describes the diffusion of a cultural innovation through a Culturally Effective Population (CEP) [12] i.e. a relatively closed network of communicating individuals or subpopulations. The way in which an innovation becomes a stable part of a community's cultural repertoire is treated as analogous to the way a disease becomes endemic in a population [30]

A formal derivation of the model is given in S1 Appendix. Briefly, consider the emergence of rock art in a metapopulation comprising N subpopulations or communities, where *I* and *S* are respectively the proportion of *infected* communities (communities that have acquired the ability to produce rock art), and *susceptible* communities (communities that have not). The innovation is transmitted from infected to susceptible communities at rate *β*, representing the "infectivity" (attractiveness and ease of transmission) of the innovation. By analogy with empirical findings in [31], rates of encounter between communities are assumed proportional to the square root of the population density (see Results and Discussion). Infected communities *recover* (here: lose the ability or the propensity to transmit rock art-related skills) at rate *γ*. Recovery may be due to death of the individuals with the requisite skills, community destruction or community fission. As envisaged in [12,13], small communities, where only a few individuals possess a given skill, are particularly vulnerable to such losses.

In this model, the rate of change in the proportion of infected subpopulations is given by:

$$\frac{dI}{dt} = \beta \text{IS} - \gamma \text{I}$$

Assuming each subpopulation is in contact with *kS* susceptible sub-populations, and that rates of encounter between communities are proportional to the square root of the population

density $\rho$, it can be shown (see S1 Appendix) that there is a critical population density, determined by the ratio of the rate of recovery, $\gamma$ to the rate of transmission, $\beta$:

$$\rho^* = \left(\frac{\gamma}{\beta}\right)^2$$

Below $\rho^*$, no communities are permanently "infected" with the innovation. Above $\rho^*$, a growing proportion of communities are infected (see Fig 1). In these conditions, the innovation is *endemic*: even when a subpopulation loses the innovation it can reacquire it from other subpopulations where it has been retained. The analogy with disease is clear. Examples of the loss and subsequent reacquisition of lost technologies are well-attested in the ethnographic literature (for example the reported loss and reintroduction of kayak-building skills among the Iniguit people [32,33]).

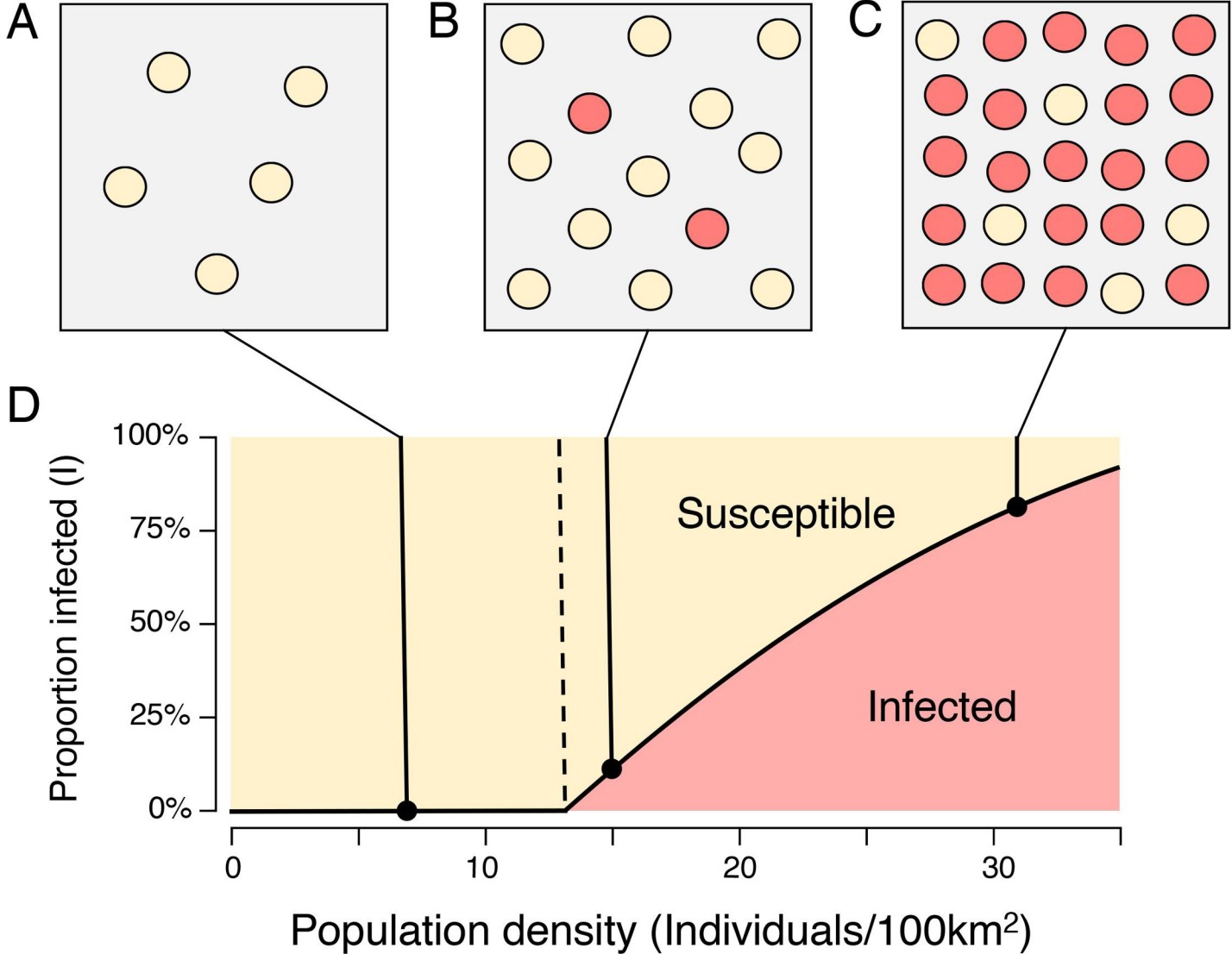

**Fig 1. The epidemiological model.** As population density increases, opportunities for transmission between subpopulations also increase. **A:** Below the critical threshold $\rho^*$ the innovation is rapidly extinguished. **B-C:** Above the critical threshold, the proportion of infected subpopulations is an increasing function of population density.

The proportion of infected subpopulations in a population is not directly observable in the archaeological record. To test our model, we therefore consider the presence or absence of rock art in "cells", each defined by its position in a two-dimensional lattice of equally sized hexagons covering the surface of the globe, each approximately 100 km wide, and each associated with a 25-year time window. Each cell is associated with a population density (Methods). Cells where the first recorded appearance of rock art falls within the time window for the cell are defined as *sites*. The *site detection ratio*, *P*, the ratio between the number of sites and the number of non-sites, is given by the expression:

$$P = (1 - \varepsilon)\zeta I^* \rho + \varepsilon$$

where $I^*$ is the proportion of infected subpopulations when $\rho = \rho^*$, $\zeta$ represents the joint effects of geology, climate and research effort (see S1 Appendix) and $\varepsilon$ is an error term (see Methods). The model predicts that for $\rho \leq \rho^*$, the detection ratio will be close to zero and that at higher densities it will rise in approximately direct proportion to $\rho$. In sum, the epidemiological model predicts, like other demographic models, that *cells with higher population will produce more rock art*. Thus, the distribution of population densities for cells containing rock art will differ significantly from the distribution for non-sites (nearly all cells) and the median population density for sites will be higher. Unlike other models, the epidemiological model also predicts a *threshold effect*: in cells whose population density is below the critical threshold, the frequency of rock art will be close to zero, rising only when population density exceeds the threshold.

## Results and discussion

### Initial analysis

To test the predictions of the epidemiological model, we combined a dataset of 133 scientifically dated rock art sites, with estimated population densities from the population model described above (see Fig 2A, 2B and S1 Table, Methods). All except 5 of these records referred to sites located between 20˚- 60˚N and 10˚ - 40˚S and with dates more recent than 46,300 years ago. The most relevant comparison was thus with all cells *in this range of latitudes and dates*. We therefore filtered the initial data to include only cells lying in this range. To avoid the facile inference that sites with no population produce no rock art, cells with inferred population densities of zero, were also excluded. After filtering, the final dataset included 9.8 million cells and 119 sites.

After inferring the population density of each cell (Methods), we compared population densities for *sites* and *non-sites*. As predicted, the two distributions differed significantly (two sample KS test D = 0.58, p<0.0001). The proportion of sites in cells with low population densities was much lower than the proportion of non-sites, and the proportion in cells with high densities was much higher (median population densities: Sites: 29.77/100km$^2$; non-sites 15.29/100km$^2$, Mann-Whitney U = 327, p<0.01) (See Table 1A). These findings are compatible with our model but also with other models where locations with higher populations produce more rock art.

To test the specific predictions of the epidemiological model, we used a Bayesian statistical framework to estimate the most likely parameter values for the model, given the empirical observations (Fig 3A, Methods). Posterior distributions for model parameters were tightly constrained (see S1 Fig). The inferred critical population density for the model with the highest likelihood, was 12.27 individuals/100km$^2$ (95% CI: 7.36–16.64) (Fig 3B). Comparison of the model against a null model (equal site frequency at all population densities) strongly supported rejection of the alternative model. Comparison against a proportional model, where detection

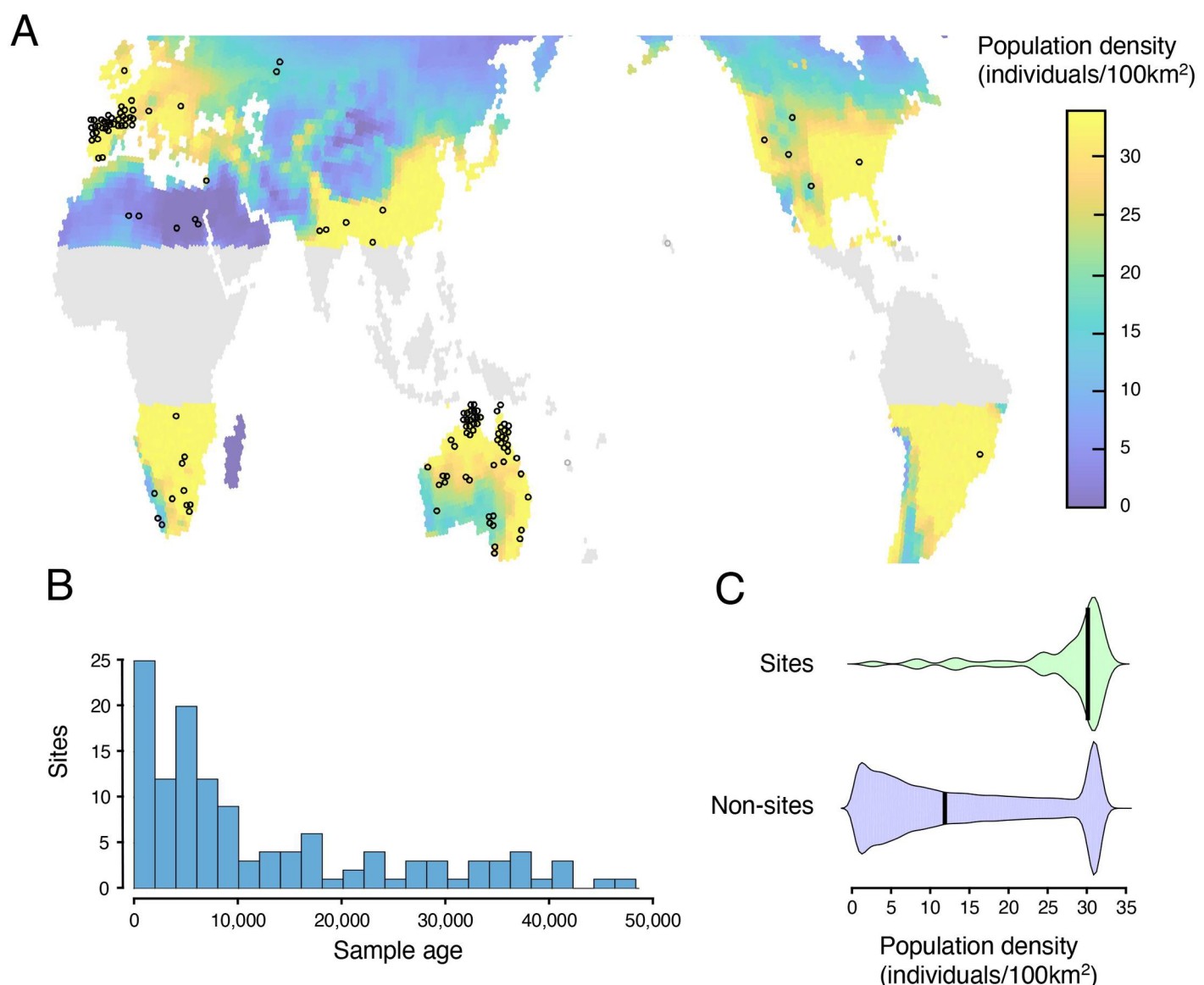

**Fig 2. Sites used in the analysis. A:** Geographical distribution of sites (all 133 sites) and inferred population distributions (maximum value over the last 46,300 years, see color bar for scale; areas excluded from the analysis shown in grey). **B:** Distribution by earliest date of rock art at site location (119 sites). **C:** Comparison of population densities for sites (density at date of first recorded rock art) vs. non-sites.

**Table 1. Summary of statistical results for the analyses described in the text.**

| Analysis | Eriksson | Australia | France-Spain-Portugal | Rest of the world | 0–9,999 years ago | 10,000–46,300 years ago | Eriksson exact direct dates | Including sites with zero inferred population | Timmermann | Eriksson φ = 1.5 | Eriksson φ = 2.5 | Eriksson φ = 6.0 |
|---|---|---|---|---|---|---|---|---|---|---|---|---|
| | (A) | (B) | (C) | (D) | (E) | (F) | (G) | (H) | (I) | (J) | (K) | (L) |
| N sites | 119 | 56 | 35 | 28 | 74 | 45 | 37 | 127 | 93 | 119 | 119 | 119 |
| N. cells | 9,817,159 | 1,223,500 | 240,156 | 8,351,650 | 2,672,852 | 7,144,307 | 9,817,159 | 10,178,765 | 213,207 | 9,817,159 | 9,817,159 | 9,817,159 |
| Median population density sites | 29.77 | 30.28 | 28.46 | 26.07 | 30.34 | 27.04 | 30.3 | 29.53 | 25.04 | 29.77 | 29.77 | 29.77 |

*(Continued)*

**Table 1.** (Continued)

| Analysis | Eriksson | Australia | France-Spain-Portugal | Rest of the world | 0–9,999 years ago | 10,000–46,300 years ago | Eriksson exact direct dates | Including sites with zero inferred population | Timmermann | Eriksson φ = 1.5 | Eriksson φ = 2.5 | Eriksson φ = 6.0 |
|---|---|---|---|---|---|---|---|---|---|---|---|---|
| Median population density non-sites | 15.29 | 21.5 | 26.74 | 13.37 | 19.92 | 13.65 | 15.29 | 14.55 | 18.04 | 15.29 | 15.29 | 15.29 |
| KS D | 0.58 | 0.64 | 0.68 | 0.55 | 0.76 | 0.45 | 0.73 | 0.56 | 0.54 | 0.58 | 0.58 | 0.58 |
| KS p | 0.000016 | 0.000001 | 0 | 0.0000053 | 0 | 0.000132 | 0 | 0.000023 | 0.000336 | 0.000016 | 0.000016 | 0.000016 |
| Mann-Whitney U | 327 | 245 | 260 | 391 | 205 | 403 | 241 | 365 | 245 | 327 | 327 | 327 |
| Mann-Whitney p | 0.0026 | 0.000048 | 0.00079 | 0.023526 | 0.000005 | 0.034209 | 0.000034 | 0.0046 | 0.007992 | 0.0026 | 0.0026 | 0.0026 |
| Threshold CI 0.025 | 7.36 | 11.84 | 0.25 | 0.01 | 8.96 | 0.92 | 9.64 | 9.53 | 12.8 | 7.86 | 6.54 | 5.27 |
| Threshold CI 0.25 | 10.72 | 17.3 | 10.32 | 1.24 | 13.18 | 4.26 | 13.76 | 12.64 | 15.64 | 11.04 | 10.03 | 9.45 |
| Threshold CI 0.5 | 12.27 | 26.81 | 16.4 | 4.55 | 15.03 | 6.83 | 15.41 | 14.28 | 17.19 | 12.67 | 11.55 | 11.03 |
| Threshold CI 0.75 | 14.05 | 28 | 19.25 | 9.7 | 16.53 | 9.53 | 16.77 | 15.66 | 18.94 | 14.45 | 13.35 | 12.78 |
| Threshold CI 0.975 | 16.64 | 28.9 | 30.04 | 17.8 | 18.82 | 14.53 | 18.65 | 17.49 | 21.79 | 16.76 | 16.17 | 15.82 |
| Log Likelihood | -1404 | -593 | -337 | -374 | -820 | -560 | -467 | -1508 | -791 | -1404 | -1404 | -1404 |
| AIC epidemiological model | 2814 | 1191 | 681 | 753.68 | 1646 | 1127 | 941 | 3023 | 1588 | 2816 | 2816 | 2816 |
| AIC proportional model | 2835 | 1208 | 686 | 751.69 | 1660 | 1132 | 963 | 3572 | 1602 | 2835 | 2835 | 2835 |
| AIC null model | 2934 | 1233 | 690 | 763.94 | 1703 | 1170 | 1000 | 3124 | 1627 | 2934 | 2934 | 2934 |
| Bayes Factor (w.r.t. proportional model) | 1.07E+05 | 1.47E+04 | 4.89E+01 | 1 | 3.88E+03 | 3.49E+01 | 1.54E+05 | 3.99E+199 | 2.55E+03 | 1.06E+05 | 9.82E+04 | 1.03E+05 |
| Bayes Factor (w.r.t. null model) | 9.79E+26 | 8.72E+09 | 9.92E+02 | 1.25E+03 | 2.29E+13 | 1.50E+10 | 6.33E+13 | 5.54E+22 | 2.22E+09 | 9.71E+26 | 8.97E+26 | 9.43E+26 |
| N. sites with sub-threshold population | 6 | 17 | 0 | 1 | 5 | 1 | 0 | 15 | 18 | 6 | 5 | 5 |
| Proportion of non-sites with sub-threshold population | 0.42 | 0.66 | 0.05 | 0.19 | 0.39 | 0.3 | 0.5 | 0.48 | 0.46 | 0.43 | 0.39 | 0.39 |
| Probability of obtaining N or fewer sites (binomial distribution) | 1.78E-20 | 3.51E-08 | 0.183 | 0.02 | 2.87E-10 | 2.30E-06 | 5.93E-12 | 3.17E-18 | 4.49–08 | 2.74E-21 | 3.79E-19 | 3.79E-19 |

**Table 1.** The Kolmogorov-Smirnov test tests the null hypothesis that the distribution of population densities for *sites* does not differ significantly from the distribution for *non-sites*. The Mann-Whitney test (which is less sensitive) tests for differences between the medians. Threshold CIs show the inferred confidence intervals for the threshold, given the empirical data. Log likelihood shows the log likelihood of the model, given the empirical data. The two Bayes factors show the ratio of the marginal likelihood of the epidemiological model to the marginal likelihood of alternative models (see Methods). The data for AICs (Akaike information criterion) [34] provide a measure of goodness of fit, taking account of the numbers of parameters in each model. The final three lines show the total number of sites with populations densities below the inferred critical threshold, the proportion of non-sites with population density below the critical threshold, and the probability of obtaining the observed number of sites or fewer, if measurements for sites and non-sites were independent and drawn from the same distribution (binomial distribution).

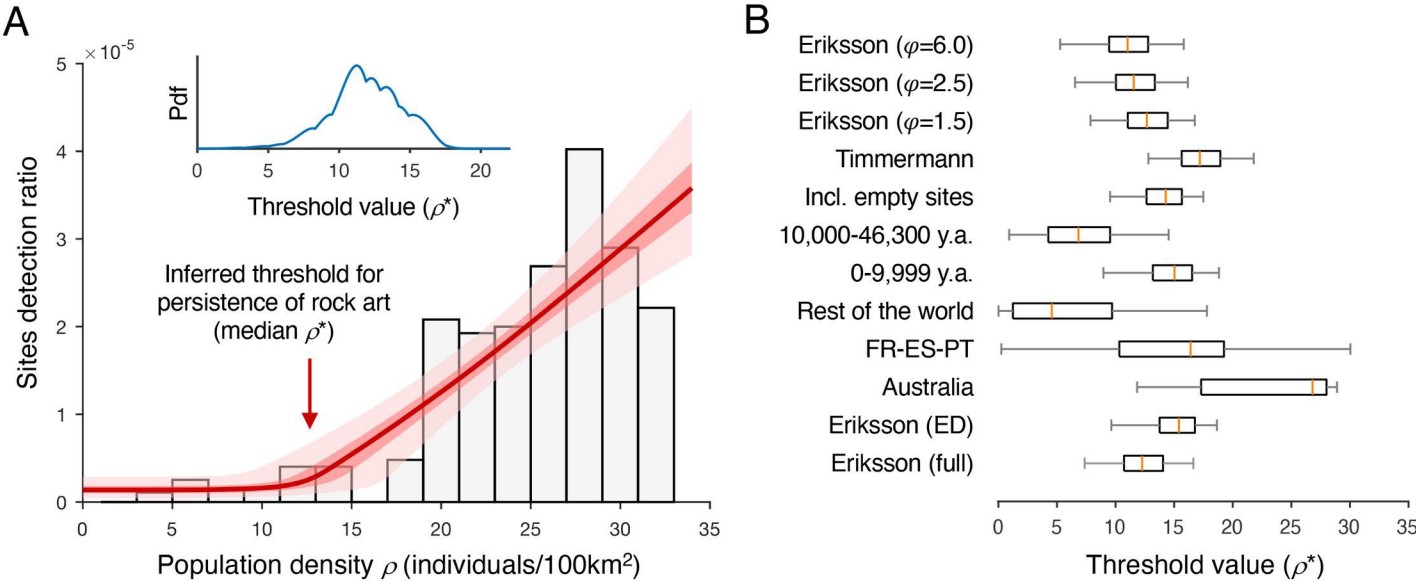

**Fig 3. Empirical support for the epidemiological model.** A. Inferred detection ratios given the epidemiological model, the full archaeological dataset, and the combined population estimates from [35,36] (see Methods). Grey bars: Empirical rock detection ratios for different intervals of population density. Red line: Estimated rock art detection ratio as a function of population density (median of posterior distribution of estimated detection ratio, with interquartile range in dark pink and 95% CI in pale pink). The red arrow on the main axis shows the most likely values of the inferred threshold, i.e., the median of the posterior probability density function (pdf) shown in the inset). B. Inferred values of the critical threshold for the different analyses described in the paper (FR-ES-PT: France-Spain-Portugal; ED: exact direct; orange line: maximum likelihood estimate; box: interquartile range; whiskers: CI 0.025–0.975).

ratios were directly proportional to population density (i.e. to the number of potential inventors in the population [11]), again supported the superiority of the epidemiological model (Table 1A). The value of the Akaike Information criterion [34] was lower for the epidemiological model than for the other two models. (Table 1A). This is evidence that the superior fit of the epidemiological model was not due to complexity (3 parameters as opposed to 2 in the proportional model and 1 in the null model).

As an additional test, we compared the cumulative frequency distribution of sites and non-sites (S2A Fig). The results (Table 1A) show that **6/119** sites (5.0%) had inferred population densities below the inferred critical threshold whereas the equivalent proportion of non-sites with population densities in the same range (44.2%) was much higher. If sites were independently distributed, the cumulative probability of finding 6 or fewer sites would be vanishingly small. Even considering some degree of dependency between values for sites in the same geographical area, the difference between the two cumulated distributions is extremely large. Taken together, these findings support the hypothesis that endemic production of rock art requires population density above a critical value.

## Sample bias and missing data

The results of the analysis reported above make it unlikely that the observed relationship between population density and the frequency of rock art sites is due to noise in the data. However, these findings do not exclude the possibility of bias, a risk aggravated by the sparse nature of the dataset.

A comprehensive survey of the vast rock art literature was beyond the scope of our study. Moreover, ecological factors make it likely that even the full archaeological record represents only a tiny fraction of the total production of rock art over the last 46,000 years. Geographical variations in research effort and the quality of research [37] add further distortions to the

picture. Probably as a result of these factors, large geographical areas in Central/South America, Central Africa, East and South-East Asia are practically unrepresented in our dataset (see Fig 2A). Furthermore, 60.1% of the sites in the dataset are less than 10,000 years old, and only 3.9% date from before 40,000 years ago (see Fig 2C).

To test the potential impact of these imbalances, we generated five filtered datasets representing three geographical areas (the territory covered by the modern countries of France, Spain and Portugal; Australia; and the "Rest of the World"), and two distinct periods (all sites with dates from the present until 9999 years ago, and all sites with dates from 10,000 to 46,300 years ago (see Methods). We then tested whether the relationship between population density and the frequency of rock art in the filtered datasets was qualitatively similar to the relationship observed in the original dataset.

All five analyses (Table 1B–1F) found different density distributions for sites and non-sites (S2B-S2F Fig), as well as higher median population densities for sites and non-sites. Three out of five found stronger support for the epidemiological model than for the alternative models and a statistically significant deficit in cells with population densities below the inferred critical threshold. The analysis for France-Spain-Portugal showed weak support for the epidemiological model but only because very few cells (sites and non-sites) had low inferred population densities. The analysis for "the Rest of the World" showed approximately equivalent support for the epidemiological and proportional models but less support for the null model. Notably, the absolute level of population density for rock art sites for "France-Spain-Portugal" and for "the Rest of the World" was in the same high range observed for the other analyses (Fig 3B, Table 1B–1F for details of these analyses). Taken together, these findings provide additional evidence in favor of the epidemiological model. Nonetheless, no analysis can ever identify or control for all possible factors capable of causing over- or under- representation of rock art sites with particular population densities. Many regions where research effort has been historically weak (e.g., equatorial Africa, east and south-east Asia, northern South America and Central America) have high ancient population densities but in others (e.g., northern Asia and the northern part of North America) densities are low. It is impossible to predict whether future discoveries will strengthen or weaken support for the epidemiological model.

## Sensitivity to dating errors

Five sites in the rock art dataset are located in cells with estimated population densities below the critical value inferred from the model, directly contradicting its predictions. We hypothesized that these findings could be due to dating errors.

Some kinds of rock art (e.g., petroglyphs) do not use organic pigments capable of providing an exact date for the artifact. In these cases, dating relies on the analysis of overlying and underlying materials providing minimum and/or maximum dates or on the analysis of organic materials stratigraphically associated with the artifact. Dates obtained in this way may be distant from the actual date of the artifact. A further complication comes from studies with radiocarbon dates whose calibration status is not explicitly stated by the original authors and whose dating is thus inherently ambiguous. It is possible, furthermore, that some uncalibrated dates, particularly from older studies, may be of poor quality.

Correct inference of population densities requires correct dates. We therefore repeated our analysis using only sites with "high quality" dates, i.e., exact dates obtained with direct methods, and, in the case of radiocarbon dates, dates calibrated by the original authors. In this new dataset, all sites (37/37) had population densities higher than the inferred critical population density. This finding matches the predictions of the model and supports the hypothesis that the unexpectedly high inferred population densities for some of the sites in the original dataset

may have been due to erroneous dating. As in the original analysis, the distributions of population densities for sites differed significantly from the distribution for all cells (Two sample KS test D = 0.73, p<0.00001) (S2G Fig). Median population density for sites was higher than for non-sites (Sites: 30.30; Non-Sites 15.29, Mann-Whitney U = 241, p<0.0001) and empirical support for the epidemiological model was much stronger than for alternative models. The inferred critical population density (15.41 individuals/100km$^2$, 95% CI: 9.64–18.65) overlapped with the estimate from the original analysis (see Fig 3B, Table 1G).

## Impact of excluding sites with inferred populations of zero

The original analysis deliberately excluded cells with inferred population densities of zero (see above). However, this procedure eliminated eight sites in our rock art dataset whose population densities were estimated incorrectly due to inaccuracies in the population model (e.g., the identification of coastal sites as "sea", failure to take proper account of the role of the Nile on Egyptian populations, issues with the timing of human expansion into South America). Given that the actual population density of these cells was probably very low (several were in desert locations) their exclusion could have biased the analysis in favor of the epidemiological model. To test this possibility, we repeated the analysis including these sites. The results (see Table 1H, S2H Fig) showed that their exclusion had only a minimal impact on the findings.

## Alternative population estimates

To control for possible inaccuracies in our population model, we repeated our analyses using more recent population estimates from Axel Timmermann [38]. These estimates were based on a different climate model and different assumptions than our original model (see Methods). As in our earlier analysis, we found significant differences between the population density distributions for sites and non-sites, (two sample KS-test D = 0.54, p<0.001) (S2I Fig). Median inferred population was higher for sites than for non-sites (sites: 25.04; non-sites: 18.04, Mann-Whitney U = 236, p<0.01) and empirical support for the epidemiological model was much stronger than for the alternative null or proportional models. The inferred critical population density (17.19 individuals/100km$^2$, 95% CI: 12.8–21.79) overlapped with the CI for the original analysis. As in the original analysis, the number of sites in cells with population densities below the threshold was significantly lower than expected if sites and non-sites shared the same distribution (see Table 1I, Fig 3B).

## Population density as a proxy for social contacts

Another potential problem was the use of population density as a proxy for contacts between subpopulations. Grove proposes that individual encounter rates depend not just on population density but on the product of density and mobility and that these variables are inversely related [31]. This suggests the theoretical possibility that high mobility, low density, populations could achieve encounter rates comparable to those of higher density populations and, by implication, a comparable ability to incorporate innovations in their cultural repertoire. Empirical results in the same paper suggest that the mobility of modern hunter gatherers is inversely proportional to the square root of population density and thus that encounter-rates are directly proportional to the square root of population density. Our model extends this assumption to inter-community encounters in ancient populations. We recognize, however, that the evidence supporting this assumption is relatively weak.

To test its validity, we defined a generalized model in which encounter rates are proportional to population density raised to the power $\frac{1}{\varphi}$ (see Methods). Fitting this model to our rock art dataset, we found that large changes in the value of $\varphi$ lead to relatively small changes in the

value of the inferred critical threshold. Although the Bayes factors with respect to the proportional and null models were lower than with our original choice of $\varphi = 2$, the differences were small (Table 1J–1L). In other words, the empirical evidence supports the validity of the epidemiological model, independently of specific assumptions concerning the exact quantitative relationships between encounter rate and population density. The robustness of our findings for the cases of rock art, makes it very unlikely that high population mobility can *completely* compensate the negative effects of low population density, at least for this class of artifact. It is nonetheless likely that high mobility and long-distance social contacts can compensate for *some* of these effects, creating Culturally Effective Populations that are much larger than the census population of a specific area [19].

## Conclusions

This study models one of the key mechanisms required for CCE (the process whereby an innovation becomes endemic in a population), and tests the predictions of the model for the case of parietal rock art. In all the analyses presented above, the distribution of population densities for sites differs significantly from the distribution for non-sites and median population densities for sites (25–30 individuals/km$^2$) are consistently higher. These results provide robust grounds to reject the null hypothesis that the emergence of rock art is unrelated to demography, and to support the general notion that "places with more people produce more rock-art".

In our main analyses and in all except two of our partial analyses, the epidemiological model has stronger empirical support than alternative proportional or null models, even taking account of the additional complexity of the model. In these analyses, the number of rock art sites with population densities below the inferred critical threshold is many times lower than would be expected from the frequency distribution of non-sites. Furthermore, the median population density for rock sites is consistently high, even when the population density of non-sites is also high. While the analysis cannot exclude all possible sources of bias, these findings suggest that population density above a critical value is indeed a *necessary* condition for rock art to become endemic in a population. The final determination of the contribution of population density to the emergence of rock art will depend on future discoveries and on data from known sites that have yet to be fully investigated.

Importantly, nothing in our model or empirical results suggests that a minimum level of population density is a *sufficient* condition for rock art. Indeed, many areas of the globe with high inferred population densities for the relevant periods (e.g., equatorial Africa), have little or no reported rock art. It should be added that the epidemiological model is not the only possible explanation for the threshold effect we observe. In reality, the mechanism described by the model represents just one aspect of CCE–namely, the process whereby an innovation becomes endemic in a population. It should thus be considered as complementary to other demographic models reviewed in the Introduction.

Previous demographic models have posited a relationship between cultural evolution and the size of so-called *Culturally Effective Populations*, networks of cultural exchange spanning potentially large geographical areas. The theoretical arguments for this relationship are strong. However, in archaeological contexts where local communities' cultural links to other communities are unknown, the size of Culturally Effective Populations is hard to estimate [12,19]. One of the key methodological features of our study is the use of population density as the independent variable. Since population densities are easier to estimate than the sizes of Culturally Effective Populations sizes, this greatly facilitates the process of theory testing.

A second key feature is the choice of "detection ratios" as the dependent variable. In general, previous empirical studies have focused on the "cultural complexity" of broad classes of

technology (e.g., food gathering technology). usually represented by the size of the relevant toolkit [39]. Here, by contrast, we test the ability of a model to predict the global, spatiotemporal distribution of a narrowly defined class of artifact (parietal rock art) over a period of 46,300 years This approach circumvents the need to operationalize the concept of cultural complexity in archaeological settings, where complete toolkits are rarely available. Taken together, these methodological features of our study will facilitate the application of our model and methods outside the case of rock art and in contexts where other constructs are difficult to measure.

Our rock art dataset, population estimates, and software tools are publicly available at https://github.com/rwalker1501/cultural-epidemiology.git. Other researchers are encouraged to use them to replicate our findings, to test our hypotheses for other classes of artifact and with other population estimates, and to explore their own models.

## Methods

### Rock art dataset

To test the epidemiological model, we used a dataset of parietal rock art, generated through a literature search with Google Scholar. We are aware that the database contains only a small proportion of all rock painting sites in the world, and that it may be subject to systematic biases (see Results and Discussion)

For the purposes of our survey, parietal rock art was defined to include all figurative and non-figurative pictograms (paintings and drawings) and petroglyphs (engravings) using rock wall as a canvas. "Portable art" (e.g., figurines) and geoglyphs (i.e., large designs or motifs on the ground) were excluded from the analysis.

The survey was seeded using the query:

("rock art" OR "parietal art" OR petroglyphs OR pictographs) [AND] (radiocarbon OR AMS OR luminescence OR Uranium).

We read the top 300 articles found by the query that were accessible through the EPFL online library, together with other relevant papers, cited in these articles. Sites where drawings, paintings and engravings were reported, were systematically recorded. Sites with no radiocarbon, optical luminescence or Uranium-Thorium date were excluded.

For each dated site, we recorded the longitude and latitude of the site (where reported), its name, and the earliest and latest dates of "art" found at the site (converted to years before 1950). Where authors reported a confidence interval for dates, we used the midpoint of the confidence interval. Radiocarbon dates marked by the original authors as calibrated dates were flagged as such. The calibration status of radiocarbon dates whose calibration status was unspecified was inferred from the surrounding text and from the CIs for the dates. Dates inferred to be uncalibrated were calibrated using Calib 7.0 [40] with the IntCal 13 calibration curve. Where different authors reported different dates for a site, without disputing dates proposed by other authors, we systematically used the dates from the most recent report. We also recorded the name of the modern country where the site was located, the journal reference, the method(s) used to produce the date, the nature of the dating procedure (direct dating, indirect dating), the nature of the data provided (exact data, minimum date, maximum date, mixed), a descriptor of the artifacts found (paintings, drawings, petroglyphs etc.), and a flag showing disputed dates.

In cases where the article did not provide a latitude and longitude, online resources were used to locate the information. The main resources used were D. Zwiefelhofer's

FindLatitudeAndLongitude web site [41], which is based on Google Maps, and Austarch, a database of 14C and Luminescence Ages from Archaeological Sites in Australia, managed by A. Williams and Sean Ulm [42].

The survey generated 190 records. Records with identical latitudes and longitudes and overlapping dates were merged (5 records eliminated). Duplicate records (12), modern sites (1), sites which did not meet the inclusion criteria (13), sites where the source was deemed unreliable (5), sites where geographical coordinates were unavailable (12), sites with disputed or doubtful dates (8) and 1 site described in a retracted article were excluded. These procedures left a total of 133 records. The complete dataset is available in S1 Table.

## Estimates of population density

The population density estimates used in our study merge results from a climate-informed spatial genetic model [35] with an improved model reported in [36]. Briefly, climate estimates for the last 120,000 years, based on the Hadley Centre model HadCM3, are combined with data on patterns of modern genetic variability from the human genome diversity panel-Centre d'Etude du Polymorphisme Humain (HGDP-CEPH), and a mathematical model of local population expansion and dispersal. The model estimates maximum human carrying capacity, for each cell in a hexagonal lattice with equal area cells 100 km wide, for all dates from 120,000 years ago to the present, using time steps of 25 years. For each land cell, estimates of past precipitation and temperature are used to estimate Net Primary Productivity (NPP). No estimate is provided for cells classified as sea. Maximum human carrying capacity is estimated by a continuous function of NPP governed by two NPP threshold values and a maximum carrying capacity parameter, K. The carrying capacity is zero below the lower NPP threshold, increases linearly from zero to K between the two NPP thresholds, and is constant and equal to K for NPP above the upper NPP threshold value. Human expansion out of Africa is simulated using a model where populations that have reached the maximum carrying capacity of a cell expand into neighboring cells. Approximate Bayesian Computing is used to compare models with a broad range of parameter values and starting values. Model evaluation is based on their ability to predict regional estimates of pairwise time to most recent common ancestor (TRMCA). Population density estimates for Euroasia, Africa and Australia were computed using parameters from the high-ranking scenario described in Fig 2 and Movie S1 in [35]. The estimates for the Americas (considered as a single continent) were taken from [36]. Compared to the model presented in [35], this paper contains a more accurate model of ice sheet dynamics in the North of the American continent and related areas of Eastern Asia, more accurate estimates of NPP across the Americas and better dates for the interruption of the Beringian land bridge. Estimated dates for the colonization of the Americas are more accurate than in the original model.

These population estimates were compared against the results using a second set of population estimates reported in [38]. The data refer to the early exit scenario (Scenario A) described in the paper. As in [35], human population density estimates are based on a climate model (LOVECLIM) combined with a reaction-diffusion Human Dispersal Model. Unlike the model in [35], the estimates do not take account of genetic data. A second difference concerns the population estimator, which is based not just on NPP but also on temperature and predicted "desert faction" and incorporates *ad hoc* modeling hypotheses absent in the previous model. These include accelerated human dispersal along coastlines ("a coastal express") and a special Gaussian decay function, modeling the probability of island hopping as a function of distance. For clarity of presentation, population density estimates from the two models were converted into the same units, namely effective population/ $100km^2$.

## Representation of population densities

Cell population densities were inferred using the results from the population models described above. Cells where the earliest example of dated rock art fell within the time window for the cell were defined as *sites*. In this way, each site was associated with a single population density. This design avoided the over-representation of sites with many rock art specimens with different dates, as well as the problems deriving from correlations between the population densities associated with different specimens from the same site.

"Non-sites" were defined according to the needs of the individual analysis (Table 2). Data for sites and non-sites were binned by population density (35 bins). Numbers of sites and non-sites and detection ratios (sites: non-sites) were computed for each bin.

## Testing model predictions

To test the predictions of the model for a specific dataset, we compared the distribution of sites by population density against the equivalent distribution for non-sites. Given the very small number of sites, the number of non-sites approximated the distribution of all cells (i.e., the number of hexagons needed to span the surface of the globe multiplied by the number of time windows). Using the two-sided, two sample Kolmogorov-Smirnov test, we tested the null hypothesis that both were drawn from the same distribution. The difference between the medians of the two distributions was tested using the less sensitive Mann-Whitney U test.

Empirical support for the model was estimated using a Bayesian framework. For each of the parameters ($\gamma$, $\zeta$ and $\varepsilon$), we defined a set of possible values lying within a plausible range. Values for $\gamma$ and $\varepsilon$ were uniformly distributed between a maximum and a minimum value, values for $\zeta$ were geometrically spaced (see Table 3). We then computed the likelihood of the model for all possible combination of these values, verifying the choice of priors using the computed posterior distributions. Where the posterior probabilities associated with the lower and/or

**Table 2. Definitions of "non-sites" used in the different analyses.**

| Analysis | Definition of "non-sites" |
|---|---|
| Eriksson; Timmermann; Eriksson exact direct; Eriksson $\varphi$ = 1.5; Eriksson $\varphi$ = 2.5; Eriksson $\varphi$ = 6.0; | Non-sites with non-zero population, with dates in the range 0–46,300 years ago and locations between 60˚N and 20˚N or between 10˚S and 40˚S (Non-equatorials) |
| 0–9,999 years ago | Non-equatorials with ages between 0 and 9,999 years. |
| 10,000–46,300 years ago | Non-equatorials with ages between 10,000 and 46,300 years |
| France-Spain-Portugal | Non-equatorials in a rectangle defined by the maximum and minimum latitudes and longitudes for sites in the dataset with locations in modern France, Spain and Portugal (NW corner: lat: 51.09˚N, lon: 9.88˚W; SE corner: lat: 35.92˚N, lon 8.12˚E) |
| Australia | Non-equatorials in a rectangle defined by the maximum and minimum latitudes and longitudes for sites in the dataset with locations in modern Australia (NW corner: lat: 25.27˚S, lon: 133.77˚E; SE corner: lat: 39.16˚S, lon: 154.86E) |
| Rest of the World | Non-equatorials except cells classified as France-Spain-Portugal or Australia |
| Including cells with inferred population of zero | Non-sites (including cells with zero population), with dates in the range 0–46,300 years ago and locations between 60˚N and 20˚N or between 10˚S and 40˚S |

**Table 2:** Each of the analyses reported in the text compares the distribution of population densities for "sites" to the distribution for "non-sites", for a specific rock art dataset (see text). Here, we specify the definitions of non-sites used for the individual analyses.

**Table 3. Prior values for parameters used in the epidemiological model.**

| Parameter | $\gamma_{min}$ | $\gamma_{max}$ | $\zeta_{min}$ | $\zeta_{max}$ | $\varepsilon_{min}$ | $\varepsilon_{max}$ |
|---|---|---|---|---|---|---|
| Eriksson | 2 | 5 | 1.00E-07 | 1.00E-04 | 0 | 5.00E-06 |
| Eriksson exact direct dates | 2 | 5 | 1.00E-08 | 1.00E-02 | 0 | 2.00E-06 |
| Timmermann | 0.1 | 6 | 1.00E-07 | 1.00E-01 | 0 | 1.00E-03 |
| Australia | 3 | 6 | 1.00E-06 | 1.00E-03 | 0 | 5.00E-05 |
| France-Spain | 0.1 | 6 | 1.00E-09 | 1.00E-02 | 0 | 4.00E-04 |
| Rest of the world | 0 | 5 | 1.00E-06 | 1.00E-03 | 0 | 5.00E-06 |
| 0–9,999 years ago | 2 | 5 | 1.00E-06 | 1.00E-03 | 0 | 1.00E-06 |
| 10,000–46,300 years ago | 0.1 | 6 | 1.00E-07 | 1.00E-03 | 0 | 4.00E-06 |
| Including sites with missing data | 2 | 5 | 1.00E-07 | 1.00E-04 | 0 | 5.00E-06 |
| Eriksson φ = 1.5 | 2 | 10 | 1.00E-07 | 1.00E-04 | 0 | 5.00E-06 |
| Eriksson φ = 2.5 | 0 | 10 | 1.00E-07 | 1.00E-04 | 0 | 5.00E-06 |
| Eriksson φ = 6.0 | 0 | 2 | 1.00E-07 | 1.00E-04 | 0 | 5.00E-06 |

**Table 3:** $\gamma_{min}$, $\gamma_{max}$, $\zeta_{min}$, $\zeta_{max}$, $\varepsilon_{min}$, $\varepsilon_{max}$ represent the minimum and maximum prior values for the parameters used in the Bayesian analysis, where $\gamma$ is the rate of recovery, $\zeta$ is the probability that an infected populations will leave at one artifact in the archaeological record for a territory of known area and $\varepsilon$ is an error term.

upper bounds of the distribution were significantly greater than zero, priors were adjusted manually, and the computation repeated.

To compute the log likelihood of the model, given a particular combination of parameter values, we computed the frequency distribution of sites and non-sites predicted by the model. We then computed the log likelihood of the predicted frequencies given the observed frequencies i.e., the dot product of the observed number of sites/non-sites per bin and the logarithms of the frequencies. The log likelihood of the model was computed as the sum of the likelihoods for sites and non-sites.

Different sets of parameter values were ranked by log likelihood, making it possible to identify the set with the maximum likelihood and to infer posterior probability distributions for each of the parameters.

The value of the critical population density, $\rho^*$ was inferred inserting the most likely inferred value of $\gamma$ in the model:

$$\rho^* = \gamma^2$$

The most likely model was then compared against the most likely proportional model, $z(\rho) = \zeta\rho$, the most likely constant model, $z(\rho) = k$. Parameter values for these models were inferred, using the same methods used for the epidemiological model. The relative likelihoods of the models were quantified using Bayes factors. We also computed the Akaike Information Criterion (AIC), a measurement of fit, which attempts to account for the number of parameters used to estimate the model (models with fewer parameters receive a higher score). Finally, we counted the number of cells with population densities below the inferred critical threshold, compared this number against proportion of non-cells with population densities in the same range, and used the binomial distribution to compute the probability of obtaining the observed or a lesser number of sites, under the assumption that the sites and non-sites belonged to the same distribution. The results of these computations are shown in Table 1.

## Sensitivity to the relationship between population density and encounter rate

To test the sensitivity of our model to the assumed inverse quadratic relationship between population density and encounter rate, we formulated a generalized version, where the critical

population density and the encounter rates are power functions of population density. Thus:

$$\rho^* = \gamma^\varphi$$

$$I^* = \begin{cases} 0, \rho \leq \rho^* \\ 1 - \left(\dfrac{\rho^*}{\rho}\right)^{\frac{1}{\varphi}}, \rho > \rho^* \end{cases}$$

The results of our analysis with this generalized model (see Table 1J–1L) are discussed in the text.

## Software

Data extraction and analysis was based on custom code written in Python 2.7, using the Anaconda development environment. Python source code for the software used to perform the analyses is available under a GPL 3.0 license, at https://github.com/rwalker1501/cultural-epidemiology.git. The software includes: (i) the script used to generate the figures and tables shown in this paper; (ii) methods to run additional data analyses and to produce figures not shown in the paper; (iii) a menu driven program providing easy to use access to these functions; (iv) documented source code for the statistical calculations and plots used in the paper and the additional analyses.

## Supporting information

**S1 Appendix. Derivation of the epidemiological model.**
(PDF)

**S1 Table. Rock art dataset (133 sites).**
(PDF)

**S1 Fig. Posterior parameter distributions for the epidemiological model.** Posterior distributions for the $\gamma$, $\zeta$ and $\varepsilon$ parameters for the epidemiological model as shown in Fig 3A (observed site detection rates, for the full archaeological dataset and the combined population estimates in [28,29]): A. posterior distribution for $\gamma$; B. posterior distribution for $\zeta$; C. posterior distribution for $\varepsilon$.
(PDF)

**S2 Fig. Cumulative frequency distributions for sites and non-sites.** Cumulative frequency distribution for sites (blue) and non-sites (red) The inferred critical threshold ($\rho*$) is shown in green. A: Eriksson; B: Australia; C: France-Spain-Portugal; D: Rest of the World; E: 0–9999 years ago; F: 10000–46300 years ago; G: Eriksson exact direct; H: Including sites with inferred population of zero; I: Timmermann. Frequency distributions for Eriksson j = 1.5, Eriksson j = 2.5 and Eriksson j = 6.0 are identical to the distributions in A and are not shown.
(PDF)

## Acknowledgments

The idea of testing our model on the case of parietal rock art emerged from a meeting at University College, London, between Richard Walker, Stephen Shennan & Mark Thomas (University College, London) and Anders Eriksson. Axel Timmermann, International Pacific Research Center, University of Hawaii kindly contributed population estimates. Michelle

Langley, Griffith University, Brisbane, Australia; Australian National University, Canberra, Australia, contributed data from a published survey of Australian rock art. Werner van Geit, Blue Brain Project, EPFL, Switzerland, contributed important code fragments. Isabel Marquez cross-checked the data for rock art sites. Mark Thomas and Stephan Shennan, Michael Herzog (EPFL Switzerland), Ruedi Füschlin (ZHAW, Switzerland), Lenwood Heath (Virginia Tech, USA) and Francesco Walker (University of Leiden, the Netherlands) reviewed earlier versions of this manuscript, providing valuable feedback. Henry Markram, leader of EPFL's Blue Brain Project, provided vital encouragement and support.

## Author Contributions

**Conceptualization:** Richard Walker, Anders Eriksson.

**Data curation:** Richard Walker.

**Formal analysis:** Richard Walker, Anders Eriksson, Taylor Howard Newton, Francesco Casalegno.

**Investigation:** Richard Walker, Anders Eriksson.

**Methodology:** Richard Walker, Anders Eriksson.

**Software:** Richard Walker, Anders Eriksson, Camille Ruiz.

**Writing – original draft:** Richard Walker, Anders Eriksson.

**Writing – review & editing:** Richard Walker, Anders Eriksson.

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
