## [Decision Letter · Decision Letter 0]

5 Feb 2020

PONE-D-19-31833

Diffusion of cultural innovation depends on demography: testing an epidemiological model of cultural diffusion with a global dataset of rock art sites and climate-based estimates of ancient population densities.

PLOS ONE

Dear Walker,

Thank you for submitting your manuscript to PLOS ONE. After careful consideration, we have decided that your manuscript does not meet our criteria for publication and must therefore be rejected.

I am sorry that we cannot be more positive on this occasion, but hope that you appreciate the reasons for this decision.

Yours sincerely,

Peter F. Biehl, PhD

Academic Editor

PLOS ONE

Additional Editor Comments (if provided):

Your manuscript has now been seen by two referees, whose comments are appended below. You will see from these comments that especially one reviewer raised serious concerns and we cannot accept the manuscript for publication.

We hope you will find the referees' comments useful.

Reviewers' comments:

Reviewer's Responses to Questions

**Comments to the Author**

1. Is the manuscript technically sound, and do the data support the conclusions?

Reviewer #1: Yes

Reviewer #2: No

2. Has the statistical analysis been performed appropriately and rigorously? 

Reviewer #1: Yes

Reviewer #2: No

3. Have the authors made all data underlying the findings in their manuscript fully available?

Reviewer #1: Yes

Reviewer #2: Yes

4. Is the manuscript presented in an intelligible fashion and written in standard English?

Reviewer #1: Yes

Reviewer #2: Yes

5. Review Comments to the Author

Reviewer #1: This is a very interesting paper that, for the most part, is concisely and clearly written. My comments are mostly minor issues dealing with terminology and grammar. The methods are well-defined and the treatment of data (especially that of the spatial data) is transparent and responsible. However, I do have a substantial concern with the authors' approach to radiocarbon data that should at least be addressed in the text. I recommend accepting this paper, pending minor revisions.

Line 98: The term "globals" comes off initially as very strange and confused me for some time. Spend another sentence explaining that you are defining cells in your raster data set based on their intersection with your data set of archaeological sites. Adding "cells" (global cells, site cells) would also help disambiguate the terms "globals" and "sites."

Line 283: Change to "site dates" (proper adjectival form) or otherwise clarify grammar.

Line 295: The term in English is mobiliary art. Avoid unnecessary foreign-language terms.

Lines 302-308: This is a very mixed bag of dating procedures that presents some serious issues with interoperability. I would not expect a meta-analysis to reinvent the wheel in this regard, but you should expound a little bit more on how dates were handled. Did you use any calibrations/corrections (14C yrs BP versus cal BP presents a big difference)? How did you handle varying uncertainty terms? Or did you take all reported dates at face value?

Addendum: Checking the supplement, I see that not all of these 14C dates are calibrated. Comparing calibrated and uncalibrated dates, depending on what portion of the calibration curve you're dealing with, is like comparing apples and oranges. The best practice here would be to recalibrate all of the dates yourselves, or to exclude all uncalibrated dates. On the sort of very general temporal scale that you are dealing with, I don't think this necessitates rerunning your analysis. However it potentially causes serious issues with the chronological assignments of some of your sites that the reader should be made aware of.

Lines 324-327: This is a long and confusing sentence. Please clarify.

Line 353: Can you be more specific? North American? Laurentide Ice Sheet? The Americas are a big place.

Reviewer #2: This is an innovative and interesting effort to test a model that uses population size to consider the process of uptake or rejection of the spread of specific cultural practices. Whilst the construction of the model and its testing is interesting, I am sceptical that such an approach can work for anything but the most basic utilitarian cultural traits because the size of population needed to borrow more complex aspects of symbolic culture need not extend beyond family level as ethnographic examples from Khoisan and Australian Aboriginal groups have shown (see exchange research by Polly Wiessner in Kalahari as an example). That said I would be interested to see this approach tried for technological developments such as early pottery, bronze and iron working. These are much more obvious cultural traits to model and with far for more sophisticated and well-dated data sets available for use. I questions why these were not tried in preference to rock art where the extent and nature of our dating is notoriously problematic? My guess is that these authors must have tried these other cultural traits first and failed.

The reason that rock art cannot serve as a test for this model is twofold: 1) there is insufficient rock art dated to know exactly when rock art appeared in any specific area; 2) our best guess at present, based on the partial data available, is that rock art arrived as part of the colonising repertoire of cultural traits in all continents except the place where it was invented: Africa. This means that rock art did not spread by a diffusionary process as is required by this model, but was brought as an embedded piece of cultural practice, hence, it cannot be used for the purposes employed here. For this reason the paper is interesting but its application flawed and I cannot recommend its publication. Were a different cultural trait substituted for rock art then this paper might become valuable.

One cautionary note is that the model assumes a single point of invention and then diffusion. It is important therefore to be able to demonstrate that the test sample was not characterised either by multiple points of invention or by migration/s. This will be hard to achieve.

6. PLOS authors have the option to publish the peer review history of their article (what does this mean?). If published, this will include your full peer review and any attached files.

Reviewer #1: No

Reviewer #2: Yes: Benjamin Smith

- - - - -

---

## [Author Response · Author response to Decision Letter 0]

8 Jun 2020

Comments from Reviewer 1

“The term "globals" comes off initially as very strange and confused me for some time. Spend another sentence explaining that you are defining cells in your raster data set based on their intersection with your data set of archaeological sites. Adding "cells" (global cells, site cells) would also help disambiguate the terms "globals" and "sites.”

We agree that our terminology here was not clear. We have thus replaced the term “globals” with the term “all cells”. Additionally, we have included a clear definition of the concept of cell that was missing in our original text (see lines 89-91, 111-115 

Change to "site dates" (proper adjectival form) or otherwise clarify grammar.

Done (see line 311)

The term in English is mobiliary art. Avoid unnecessary foreign-language terms.

We have decided to use the term “portable art” (see line 324)

“(…) you should expound a little bit more on how dates were handled. Did you use any calibrations/corrections (14C yrs BP versus cal BP presents a big difference)? How did you handle varying uncertainty terms? Or did you take all reported dates at face value?

(…) I see that not all of these 14C dates are calibrated. Comparing calibrated and uncalibrated dates, depending on what portion of the calibration curve you're dealing with, is like comparing apples and oranges. The best practice here would be to recalibrate all of the dates yourselves, or to exclude all uncalibrated dates. On the sort of very general temporal scale that you are dealing with, I don't think this necessitates rerunning your analysis. However it potentially causes serious issues with the chronological assignments of some of your sites that the reader should be made aware of.”

We thank your reviewer for emphasizing this issue. In the new version of our manuscript we have attempted to follow his suggestions. 

When testing our model, we consistently used the we considered only the oldest date for each site. Where confidence intervals were given, we used the midpoint of the interval. oldest date recorded for a site. We reviewed the calibration status of all dates used in our analysis. Where original authors did not explicitly indicate the calibration status of radiocarbon dates, we attempted to infer their calibration status from the surrounding text and from the CIs for the dates. We then calibrated the uncalibrated dates, using Calib 7.0 [41] with the IntCal 13 calibration curve. Dates we recalibrated are indicated with an asterisk in SM Table 1. We explain the way we have handled dates in lines 334-341.

We recognize that the procedure used to infer the calibration status of radiocarbon dates when this was not explicitly indicated by original authors is prone to errors. To test the sensitivity of our results to these and other potential errors in dating, we performed a new analysis in which we excluded all sites without an exact age, obtained with direct methods and excluding radiocarbon dates that the original authors had not explicitly declared to be calibrated. The results of the analysis with this restricted dataset, (see Table 1) were very similar to those obtained with the full dataset. We describe the new study in 166-186 and 430-442.

This is a long and confusing sentence. Please clarify.

We agree with your reviewer and have rewritten the sentence (see lines 359-362)

“Can you be more specific? North American? Laurentide Ice Sheet? The Americas are a big place”

In our analysis, the original population density estimates for the North and South American continents contained in Eriksson 2012, were completely replaced by revised estimates from Raghaven, 2015 (see references). My co-first author, Anders Eriksson, was an author on both papers. The reasons for the replacement are explained in lines 379-384 of the manuscript 

Reviewer 2

Reviewer 2’s negative evaluation may be partially due to the “diffusionist” language we use to describe and interpret our model. To avoid misunderstanding, we have changed the title of our paper, and clarified the description and interpretation of our model. In what follows we respond to Reviewer 2’s evaluation in greater detail.

The paper is not technically sound. The statistical analysis was not performed appropriately or rigorously 

Reviewer 2 states that our work is not technically sound and denies that our statistical analysis was performed appropriately or rigorously. Unfortunately, he does not motivate these evaluations. His comments raise no issues about our sampling methods, rock art data, choice of controls, previously published population estimates, sample size, checks for confounders, or use of different datasets to test our model. It is thus difficult for us to respond.

Reviewer 2’s main argument is his skepticism that

 “... such an approach can work for anything but the most basic utilitarian cultural traits because the size of population needed to borrow more complex aspects of symbolic culture need not extend beyond family level as ethnographic examples from Khoisan and Australian Aboriginal groups have shown (see exchange research by Polly Wiessner in Kalahari as an example).”

He goes on to explain that

 (...) the reason that rock art cannot serve as a test for this model is twofold: 1) there is insufficient rock art dated to know exactly when rock art appeared in any specific area; 2) our best guess at present, based on the partial data available, is that rock art arrived as part of the colonising repertoire of cultural traits in all continents except the place where it was invented: Africa. This means that rock art did not spread by a diffusionary process as is required by this model, but was brought as an embedded piece of cultural practice, hence, it cannot be used for the purposes employed here

These objections appear to be based on misunderstandings of our model that we will discuss below. More importantly, they represent nothing more than the reviewer’s personal opinion. Even if this opinion were widely shared, such a consensus would not be a valid reason to refuse engagement with our data and analytical results.

The data we present provide overwhelming support for a relationship between population density and the presence of rock art, and strong support for the specific predictions of our model. In these circumstances, a critical reviewer would normally seek to attack the results (e.g. by demonstrating systematic biases or cherry-picking of data), the analysis (e.g. by criticizing the choice of statistical methods), or the interpretation of the results (e.g. by suggesting alternative explanations for empirical findings). Reviewer 2 does not offer this kind of critique.

Unsubstantiated insinuation

Reviewer 2 writes 

“I question why these [other technologies] were not tried in preference to rock art where the extent and nature of our dating is notoriously problematic? My guess is that these authors must have tried these other cultural traits first and failed. (our emphasis)”

We would like to state emphatically that we have not hidden any negative results from previous studies (the file drawer effect). It is not acceptable that this sort of unsubstantiated insinuation should be “thrown in” in the middle of the review report.

 

Diffusionism

We recognize that our use of “diffusionist” language in the original version of our manuscript is liable to create misunderstandings. In reality, we do not make any claim about the history of rock art, globally or in any particular region. More specifically, we never claim or assume that the global distribution of rock art is the result of a process of diffusion between populations (classical diffusionism). What we actually model is the process allowing a cultural innovation to be stably maintained (i.e. to become endemic) within a metapopulation of communicating subpopulations. We have now made this point explicit in the title of our study, the abstract, and the text describing our model (see lines 57-63).

Our underlying assumption is that the ability to produce rock art, like other cultural traits, is transmitted from person to person and from subpopulation to subpopulation. Reviewer 2 posits that "rock art arrived as part of the colonising repertoire of cultural traits in all continents except the place where it was invented", If this was so, the ability to create rock art was necessarily transmitted vertically from generation to generation and horizontally between subpopulations, over very long periods of time. Furthermore, transmission must have been sufficiently reliable to ensure that the trait was never completely lost. In brief, transmission was vital. This is true regardless of the complexity or otherwise of the trait in question. 

The model we present in our manuscript, and other demographic models of cultural evolution (see references 10-16 in our paper), propose that successful transmission over many generations depends on demographics. It is possible that small social groups, with intense social contact between group members, and high local population density could maintain complex traits (e.g. rock art), even when the mean population density in the territory they inhabit is low. This is what appears to have occurred in the ethnographic examples cited by reviewer 2. However, small groups are extremely vulnerable to stochastic events in their environment and are unlikely to survive over long periods of time (see references 12-13 in our manuscript). If this is so, most such complex traits will disappear without leaving a trace in the archaeological record. In our manuscript we argue that the traits most likely to leave such traces are those that spread between subpopulations, ensuring that the cultural trait survives even when individual subpopulations are destroyed, disintegrate or lose their skills. This in turn requires a minimum level of inter-group contact. This is the reason, we argue, why most of the rock art sites in our dataset are in territories with population density beyond the critical threshold. 

The need for exact dating of rock art sites

Reviewer 2 believes that our analysis depends on exact knowledge of the dates at which rock art first appeared in a specific location. This is a misunderstanding. In reality, the only requirement for our analysis is an unbiased sample of rock art sites with accurate dates. As reported in our Methods section, we use the oldest available dates for each site (not the date for the first actual appearance of art at the site which is unknown and unknowable). This somewhat arbitrary decision was driven by technical considerations. Many excavations yield different samples of rock art with different dates. Since these cannot be treated as independent data points, we decided to use a single point for each site. We did not choose the most recent date, because in many American and Australian sites the most recent findings are modern.

Finally, we note that reviewer 2 is mistaken about the availability of reliable dates for rock art sites. Many early reports were indeed very inaccurate. However the last thirty years have seen important advances in dating methodology. In particular, artifacts produced using organic materials can now be dated directly using radiocarbon methods. Many of the reports covered by our literature survey use such methods. In a control study, presented in our paper, we repeat our analysis, restricting our rock art data to sites where exact dates had been determined using direct methods and where all radiocarbon dates have been calibrated by the original authors. We find that the likelihood of our model given this “purified” data set is higher than with the data used in our original analysis (see lines 166-186, 430-442).

We hope that in view of these considerations and of the substantial revisions we have made to our manuscript, you will be able to accept our work for publication in PLOSONE.

---

## [Decision Letter · Decision Letter 1]

16 Nov 2020

PONE-D-19-31833R1

Stabilization of cultural innovations depends on population density: testing an epidemiological model of cultural evolution against a global dataset of rock art sites and climate-based estimates of ancient population densities.

PLOS ONE

Dear Dr. Walker,

Thank you for submitting your manuscript to PLOS ONE. After careful consideration, we feel that it has merit but does not fully meet PLOS ONE’s publication criteria as it currently stands. Therefore, we invite you to submit a revised version of the manuscript that addresses the points raised during the review process.

As you can see below, the reviewers have provided substantial reports supporting publication of your manuscript. Nevertheless, they have also listed several comments, all of which should be carefully addressed in a revised version of the article. In particular, when performing such a revision, please pay special attention to the following issues:

1.- More details on several methodological aspects. Both reviewers asked for clarifications on different methodological aspects. Notice PLOS ONE's publication criterion #3: "Experiments, statistics, and other analyses are performed to a high technical standard and are described in sufficient detail."(https://journals.plos.org/plosone/s/criteria-for-publication#loc-3).

2.- Reviewer 3 required a more explicit presentation of the limitations of the work (especially those related to the nature of the data under study).

We look forward to receiving your revised manuscript.

Kind regards,

Sergi Lozano

Academic Editor

PLOS ONE

Journal Requirements:

2. We note that [Figure 2] in your submission contain [map/satellite] images which may be copyrighted. All PLOS content is published under the Creative Commons Attribution License (CC BY 4.0), which means that the manuscript, images, and Supporting Information files will be freely available online, and any third party is permitted to access, download, copy, distribute, and use these materials in any way, even commercially, with proper attribution. For these reasons, we cannot publish previously copyrighted maps or satellite images created using proprietary data, such as Google software (Google Maps, Street View, and Earth). For more information, see our copyright guidelines: http://journals.plos.org/plosone/s/licenses-and-copyright.

1. You may seek permission from the original copyright holder of Figure(s) [2] to publish the content specifically under the CC BY 4.0 license.

3. Please upload a copy of Figure 1, to which you refer in your text. If the figure is no longer to be included as part of the submission please remove all reference to it within the text.

4) Please ensure that you refer to Table 2 in your text as, if accepted, production will need this reference to link the reader to the Table.

Reviewers' comments:

Reviewer's Responses to Questions

**Comments to the Author**

1. If the authors have adequately addressed your comments raised in a previous round of review and you feel that this manuscript is now acceptable for publication, you may indicate that here to bypass the “Comments to the Author” section, enter your conflict of interest statement in the “Confidential to Editor” section, and submit your "Accept" recommendation.

Reviewer #3: (No Response)

Reviewer #4: (No Response)

2. Is the manuscript technically sound, and do the data support the conclusions?

Reviewer #3: Partly

Reviewer #4: Yes

3. Has the statistical analysis been performed appropriately and rigorously? 

Reviewer #3: I Don't Know

Reviewer #4: Yes

4. Have the authors made all data underlying the findings in their manuscript fully available?

Reviewer #3: Yes

Reviewer #4: Yes

5. Is the manuscript presented in an intelligible fashion and written in standard English?

Reviewer #3: Yes

Reviewer #4: Yes

6. Review Comments to the Author

Reviewer #3: * The introduction is good and the model captures a nice idea - that there could be a non-linear threshold of population sizes at the regional level that can sustain culture, similarly to the way an epidemic may be sustained (i.e. become endemic).

* I didn't see the full comments of reviewer 2 - only the extracts from it in the authors' letter of response - but from what appears there I agree with the authors that some of the reviewer's comments are unfair\\non-specific or reflect a misunderstanding of the modeling exercise that they carried out.

* I like the exercise, but I don't think much can come out of it in this case. I do agree with reviewer 2 that rock art data is super-sparse and not much can be stated decisively about the question of pop-size and cultural complexity/stability because of that. Moreover, crossing it with population size estimates, which are notoriously crude, makes it even worse. The bottom line is that I still think it's worth publication, but that the limitations need to be discussed and highlighted early on. See my comments below and summary at the end.

===

* I don't understand the basic thing: what is being correlated with what.

Was a single population size associated with each geographic cell (e.g. the highest population size estimate for that region over the past 46KY, or for the period 46ky-10ky, and then a separate size for 10ky to the present)? This is what the caption of Figure 2 might hint.

If not, and each site's rock art is linked to the population density estimated for the specific decade (or something else? What's the time-resolution in which population densities are estimated? This needs to be clear), then I'd expect that the background distribution of "all cells / globals" be composed of many many snapshots, i.e. if the time-resolution of the population size estimates is, say, an estimate for each 1000-year-period, then the set of all data entities that you should have in the analysis is 46 times the number of geographic regions to which you split the world. This would mean that even cells in which rock art was found should "contribute" to the dataset many instances of no-art-found, because the art in the cell is associated with one time window, not all 46.

Respectively, if things are done this way, and there are multiple dates for the same rock-art site spanning more than one 1000-year window, each of these dates should be considered, not only the oldest one.

This is the "proper" way to do this analysis, I would think, but it doesn't seem that this is what was done. If I'm understanding correctly - and I'm likely to be misunderstanding, but if I were forced to guess - it seems to me that a single population size was associated with each geographic cell (the max pop-size in the recent 46ky), and each cell received a simple yes/no with respect to rock art within the last 46ky, and that this was the basis for the main analysis (and then a second analysis did the same, but with a breakdown to two periods, before 10ky and after). Regardless of whether this guess is correct or not, this needs to be clear to the reader.

What was done in these respects needs to be clarified, and the reasoning and/or caveats of the choices made need to be explicit.

Even if what I think is the "proper" way to do the analysis differs from what you did / prefer to do that's OK - I don't want to impose my view on what's proper - but clarification of what was done, explanation of the reasoning, and discussion of the advantages/disadvantages/caveats, are crucial.

Notably, if this is what was done, it renders moot the vast majority of discussion (and annoyed-other-reviewers' comments) regarding details of dating, sensitivity to errors, and so on, because what does it matter what the precise date was, if it is then being associated with a population size estimate whose time-resolution is "max pop size within the last 46ky"?

In fact, even if a population size estimate which is more specific was used, the time-resolution of the pop-size estimates is probably the cruder factor, which make calibration errors of the carbon dating, for example, meaningless.

* In many (most?) SIR models, once an individual is in the Recovered/Removed category, they can't be re-infected (because of adaptive immunity or because of death; for transmission, the two are the same). I didn't see a mention of how/whether this is treated here. Worth mentioning.

* The model comparison is "unfair" - the epidemiological model has three fitted parameters, while the two others have zero (null) or one (the proportional model). Naturally, a model with more levels of freedom will be able to provide a better fit to the data. I didn't see how this was accounted for (some penalty function for extra parameters might make sense, perhaps).

* One of the model parameters is the rate of recovery (loss of the cultural trait of painting on rocks). Wouldn't the straight-forward way to estimate this parameter be to use the distribution of date inferences for art from the same cell? Currently it seems that you excluded from your analysis all the later dates except the oldest one?

* In truth, I don't believe the model - and I don't think anyone really does. I mean, each site of rock paintings is something produced within a single generation, or maybe 5-10 generations. 100-200 years at best; and it is being assumed as representing 36ky or 46ky; for which a single population density at the location is chosen as representative of the whole period. (If I understand correctly what was done. See above). The noise here necessarily overwhelms the signal.

Another way of saying this is: infection with a cultural trait such as rock painting is something that takes place on cultural-evolution time scales, i.e. ecological time-scales (single/few generations). But it is then correlated with pop-size estimates whose time-resolution differs by an order of magnitude (probably thousands of years? Not mentioned, it seems).

A third way of saying it, that perhaps captures an additional perspective which is important: you are fitting a model that is based on the transmission dynamics between entities on extremely short time scales - single encounters - to data for which each data-point, and each data-points' time-estimate, spans hundreds or thousands of years. That in itself is problematic. On top of that, the data is super-sparse, so even if the model is/were the precise description of what actually happened in the world, there's no chance that you'd have the events in the dataset to support it.

* Accordingly, I don't think the finding that the epiemiological model gets more support than the simpler correlation-between-art-and-pop-size is more than an artifact of the way that the comparison was done. That is, without fully understanding the Bayesian-based model-selection that you did, I'm quite certain that in the bottom line the comparison favored the more complex model because it has more levels of freedom.

* Excluding the sites in which cave art is found despite estimates of nearly zero population seems like cherry-picking; these are the strongest cases that would support a claim that there's no connection between density and presence of rock art. I don't understand the justification for throwing these out.

* In essence, and because I know pretty well how crude population size estimates are (temporally and geographically), it seems to me that what this paper is showing - using elaborate and sophisticated modeling - is that art appears in the places in which populations were generally large (not necessarily exactly at the time-point in which the art was produced; we don't know). This is sort of like saying "where there were people, there's a likelihood of finding rock paintings" (funnily, the data suggests that even this isn't correct - the 8 sites that were excluded show that art appears even where there weren't any people, according to the pop-size inference you used). From what I know about pop-size estimates and from what I think I understand you did (see above), I sincerely think that the claim can't go much further than this rendition of it.

Moreover, these locations are where archaeologists would have looked for such art as well, so there's a huge confirmation bias here. I doubt that any correction-factor that you introduced can truly account for that, regardless of how smart it is. It's an inherent problem of missing data.

===

Bottom line: I think this is a nice exercise of fitting a model to data. I don't think it's a particularly successful one.

Even though I'm personally convinced that population size does have a lot to do with cultural complexity, and that transmission dynamics are important for this, I don't think the endeavor described in this manuscript can really help to support or reject this claim.

I think it is still worth publishing, and PLOS ONE is a good venue for it, but I'd want to see the points I raised above clarified (I felt that I need to do a lot of guessing - see above - and ended up not really understanding what was done), the caveats, limitations, and alternative explanations (see my last point above) discussed, and the main claim in the abstract and those in the discussion need to be attenuated somewhat to reflect these limitations.

Sorry I can't be more supportive with respect to the manuscript as it stands. I definitely appreciate the effort that's been put into this.

Reviewer #4: In this paper the authors propose to use archaeological data retracing rock art apparition to test hypotheses about demographic changes in human population. They do so by combining a demographic model with an epidemiologic model of cultural change, to estimate the likelihood of these models given the evidences they have collected about rock art approach. The overall article is very nice, the approach is innovative and it's application to rock art and demography is unique to my knowledge. I think the paper fits perfectly within PLOS ONE aims and goals.

First of all I want to underline the great effort done by the authors to make available all the data they collected and the code used in the paper. Regarding the code, some very minor problems popped up when I first tried to use the python code from the git repository (typos in folders' name and necessary packages that were not mentioned in the readme.txt) but that can be easily corrected later on, and should not prevent the publication of the paper. The code is well documented and easy to use, which makes it, in my opinion, one of the great strength of the paper. If some of the critics made by the previous reviewers on the scarcity and uncertainty of the data may hold, the authors here offers us a way to reproduce their analysis with newer and more complete dataset once they are available.

A few points that I will detail below remain unclear to me:

1. It seems that the Figure 1 wasn't included with the publication. Its legend is given on line 66 and it's referenced on lines 62 and 79, but I could only find fig2.tiff and fig3.tiff at the end of the paper.

2. I understood that Figure 3B (and some elements of 3A) shows the values for the "threshold", which is called $\\rho^{*}$ throughout the main text, but it's reported as $r^{*}$ in this figure. I may be wrong and $r^{*}$ is something else, but then I was not able to find what it was referring to. Also, I found the left panel of this figure confusing (the A panel) : it took me a while to understand the link between all the elements represented. I think the legend of what's the red curve should appear on the figure (and not only in the caption). Here it looks like the arrow is telling us what this red line is about, which is not the case. In fact, I am not sure about including the full posterior distribution and the mention "Median threshold value" above the arrow on this same panel. Maybe the PDF of the posterior could displayed in a third panel, adding to it the median, quartiles and CIs used to calculate the value for the red curve (you could add that on the PDF as vertical lines, arrows, or a whisker plot with all these metrics, as on the panel B, below the PDF).

3. I have some questions regarding the model and the Bayesian inference :

a. The authors say: "For each of the parameters (γ, ζ and ε) we defined a uniformly distributed set of possible values lying within a plausible" it would be nice to have an idea of those values, maybe a table summarising all parameters inferred, with a brief description of what they mean and their prior distribution would help (that could even stay in the supplementary material with the posterior distribution). It would also help to have the posterior distribution of $\\rho^{*}$ in the supplementary material aside the other posteriors, as well as the joint posteriors for all parameters (as the Figure S2: in the SI of this paper: https://www.nature.com/articles/srep39122).

b. It would help also to have a few sentences explaining how the likelihood is computed. Looking at the code, it appears that a dot product is used to have a measure of the difference between the log of the estimated frequency and the real data, though I am not sure if the data is also log transformed or not (I assumed than yes). Then, I see that the frequency of sites where _no_ rock art has been found is also used to compute the log likelihood. This sounds as assuming that sites where no evidence has been found _are_ sites where rock art was absent and didn't leave any trace (though I think that it's likely that we just haven't found the evidence yet). Similar questions have been raised after the publication of the recent Nature paper by Whitehouse et al (2019) on moralizing gods and complex society. I maybe be wrong and this may be captured by the $\\epsilon$ and $\\zeta$ terms of the model, but clarifying how the log likelihoods (and thus the Bayes factors) are computed may help to interpret the results.

c. At the end, the authors mentioned the fact that they tried to run the experiment with different relations between group interaction and population density (introducing a parameter $\\phi$ to generalise their model). They say the results are the same but also that they can't show the results. Would it be possible to have them at some point? recalculating the Table 1 for, let say, 3 different values of $\\phi$ and add that to the SI should not be that hard or I am wrong?

d. Following on this interaction part of the epidemiological model, the authors also decide to "normalized" the parameter $\\beta$ to one. This sounds like a strong move to me, as if I understood correctly, this is the parameter representing the "virality" of new cultural variants, _ie_ the rate at which a cultural traits it's shared between populations. Thus, one would expect than with higher $\\beta$, the need for denser population will decreases. Then, one could argue than by setting $\\beta$ this way the authors are forcing the model to work only with denser populations, while other path to CCE may be possible. The best thing would be to infer the values of this parameter as they have done for $\\zeta$,$\\gamma$ and $\\epsilon$, but I understand than this will add complexity and I think this can be left for later studies. Nonetheless, the authors should spend a few more words to justify there choice here (maybe explaining why they think this won't change the overall outcome of the model if they think so? maybe some of the results with different $\\phi$ mentioned before may have given hint about why they can do that?).

4. Minor typos: I noticed a few typos in reference [32] "Qitdlarssuaq: Lhistoire dune migration polaire" should be "Qitdlarssuaq: l'histoire d'une migration polaire". Also on line 150 it looks like there is a problem with the parenthesis.

Aside the points mentioned above, I think this paper is a good contribution to the quantitative exploration of human cultural change. The choice of the demographic model as well as the quality, size and possible biases of the dataset could be debated endlessly. Nonetheless, the completeness of the data and the code furnished by the authors allows anyone to easily challenges, changes and compare each part of the study with his own model or data and reproduce the analysis made by the authors. This, I think, is a great step toward numerous studies that will be able to explore the heated debate on the link between Cultural Evolution and Human demography.

7. PLOS authors have the option to publish the peer review history of their article (what does this mean?). If published, this will include your full peer review and any attached files.

Reviewer #3: No

Reviewer #4: No

---

## [Author Response · Author response to Decision Letter 1]

23 Dec 2020

(please see attached "response to reviewers" for an easier to read version of this text).

Dear Mr Lozano,

We would like to thank you and your reviewers for giving us the opportunity to resubmit our manuscript entitled “Stabilization of cultural innovations depends on population density: testing an epidemiological model of cultural evolution against a global dataset of rock art sites and climate-based estimates of ancient population densities”. 

We are especially grateful to the reviewers for their thorough, careful and well-informed reading of our manuscript. Several of the comments challenged our initial assumptions, leading us to perform additional analyses, not included in the original manuscript, and to revise parts of the manuscript, where our original text was not sufficiently clear. We hope this new version of the manuscript does justice to their work. In what follows, we address comments, one by one, indicating in each case, any changes we have made to the manuscript. In each case, we discuss substantial remarks first, leaving less important issues to the end

Reviewer 3

 I don’t understand what is correlated with what. Was a single population size associated with each geographic cell (e.g. the highest population size estimate for that region over the past 46KY, or for the period 46ky-10ky, and then a separate size for 10ky to the present)?

We apologize if our original text did not make this sufficiently clear. The population densities in our analysis represents the estimated population density of a hexagonal “cell” for a specific 25 year time window. Cells where the first recorded appearance of rock art falls within the time window for the cell are defined as “sites”. The “detection ratios” used in the analysis are computed by dividing the number of sites in a bin by the total number of non-sites (measured across the complete period covered by the analysis) . We have modified the presentation of the model (lines 87-92) and our data analysis (lines 119-125) to clarify our methods.

 ... if things are done this way, and there are multiple dates for the same rock-art site spanning more than one 1000-year window, each of these dates should be considered, not only the oldest one.

Excavations at some of the sites in our dataset have produced large numbers of rock art specimens, sometimes spanning thousands of years. If we had included one date (and associated population density) per specimen (instead of one per site), we would have (i) over-represented sites with large numbers of specimens (ii) included a large number of non-independent data points (estimated population densities at different times). There is a serious risk that treating datapoints from the same site as independent would overestimate empirical support for our model. We therefore prefer to maintain the more conservative approach described in our manuscript. We have provided additional text (lines 123-127) motivating this design.

 ...each site of rock paintings is something produced within a single generation, or maybe 5-10 generations (...) but it is then correlated with pop-size estimates whose time-resolution differs by an order of magnitude (probably thousands of years? 

This objection seems to derive from the misunderstanding discussed under point 1. We agree that the parameter of interest for our model is the rate of transmission over ecological time-scales. The size of the time window in our analysis (25y) matches this time-scale. 

 ...the data is super-sparse, so even if the model is/were the precise description of what actually happened in the world, there's no chance that you'd have the events in the dataset to support it. 

We believe that a sparse sample can be informative, even if small, on two conditions: 

 The phenomenon under investigation has a sufficiently large effect 

 The sample is representative of the population i.e. the sample is not systematically biased

As concerns the first condition, our revised manuscript includes new analyses documenting the proportions of sites and non-sites with sub-threshold population densities (see Table 1) and new graphs showing their relevant frequency distributions (see SI Fig 2). In our main analysis for example, we show that just 6/119 (0.50%) sites have population densities below the inferred critical threshold, compared to 42% of non-sites. The size of the observed effect makes it extremely unlikely that our results are due to noise in the data. The latest version of our software allows users to replicate these studies for themselves.

 These locations (our note: locations with large ancient population densities) are where archaeologists would have looked for such art as well, so there's a huge confirmation bias here....

Your reviewer points to the possibility of huge bias in the archeological record of rock. We agree with this assessment and have changed the position of the relevant section in our manuscript to emphasize the importance of this issue We have also expanded and rewritten our discussion, including a new analysis to test the role of regional bias(see lines 199-235). 

The original manuscript already included analyses for the two regions in the world (France-Spain-Portugal, Australia) with the highest volume of attested rock art and two distinct periods (from 0-9,999 years ago, and from 10,000-46,300 years ago). These analyses show that the predicted effect of population density holds within as well as between regions and periods. We have now included an additional analysis using data for the remaining regions of the world. All five analyses show a strong relationship between the frequency of rock art and population density and in three out of five cases the epidemiological model is better supported than the proportional model (see lines 219-225 and the results in Table 1). The latest version of our software allows users to replicate these analyses.

To summarize, we have made a great effort to correct for the unavoidable biases present in our dataset, we believe with some success. Nonetheless, we agree with your reviewer that no correction can compensate for missing or biased data. The ultimate assessment of our model depends on results from future research. New text in the abstract (section on bias ( 26-27), in the section on bias ( 228-234) and in the conclusions (302-304) draws attention to this caveat.

 I don't think the finding that the epidemiological model gets more support than the simpler correlation-between-art-and-pop-size is more than an artifact of the way that the comparison was done. That is, without fully understanding the Bayesian-based model-selection that you did, I'm quite certain that in the bottom line the comparison favored the more complex model because it has more levels of freedom.

To clarify our procedures we have added a paragraph to the methods section, describing our Bayesian model selection procedure ( 490-506). We have also revised our text to show where the predictions of the epidemiological model coincide with those of the proportional model and where they diverge (151-174; 219-228; 293-305)

As concerns the substance of your reviewer’s objections, we still believe that there is strong evidence in favor of the epidemiological model.

 In all our analyses we compute Bayes Factors for the epidemiological model vs. the proportional and null models. In our main analysis and in three out of five of our partial analyses, the epidemiological model is many orders of magnitude more likely than alternative models.

 Our analyses show very few sites in cells with population densities below the inferred critical threshold Under the null hypothesis we would expect the proportion of sites with populations in this range to be approximately proportional to the proportion of cells. Even allowing for some dependencies in the data, the probability that the observed distribution could have arisen by chance is vanishingly small. 

 The proportional model predicts that site frequency will increase in direct proportion to population density. In our observations there is no measurable increase before population density reaches the critical threshold (see SM fig 2) . This is further evidence in favor of the epidemiological model.

 The median population density for rock art sites is in the same high range (between 25 and 30 inhabitants/100 km2 ) in all our analyses. This supports the notion that rock art occurs not just where there are more people, but where there a lot of people (in historical terms).

 As concerns the issue of complexity, the revised version of our manuscript formally accounts for differences using the Aikake Information Criterion (ref. 37) - a measure of fit that penalizes models with a higher number of parameters. The results, included in Table 1 and described in lines 164-166, 512-515, shows lower values (better performance) for the Epidemiological model than for the proportional or null models. This is evidence that our results are not due to the complexity of the model.

11) Excluding the sites in which cave art is found despite estimates of nearly zero population seems like cherry-picking; these are the strongest cases that would support a claim that there's no connection between density and presence of rock art. I don't understand the justification for throwing these out. 

We thank your reviewer for raising this potentially serious objection to our analysis. 

Our original intention in excluding “uninhabited” cells from our analysis was to avoid the facile conclusion that cells with no inhabitants produce no rock art. However, we had not considered the possibility that our population model would identify some of our sites as uninhabited at the relevant dates. 

To test the impact of excluding these sites, we have performed a new analysis which included sites and non-sites with inferred population densities of zero (. The analysis (supported in the latest version of our software) showed (to our relief!!!) that support for the epidemiological model is extremely strong even after the “uninhabited” sites are included. This result reflects the huge gap between the observed number of sites and the number expected under the null model. Even after inclusion of 8 additional sites, the gap remains extremely large. The need for this analysis and the results are described in lines 258-265

12) In many (most?) SIR models, once an individual is in the Recovered/Removed category, they can't be re-infected (because of adaptive immunity or because of death; for transmission, the two are the same). I didn't see a mention of how/whether this is treated here. Worth mentioning. 

 We have added additional text (lines 70-74) to the original text, interpreting the meaning of “recovery” in the context of the model

13) One of the model parameters is the rate of recovery (loss of the cultural trait of painting on rocks). Wouldn't the straight-forward way to estimate this parameter be to use the distribution of date inferences for art from the same cell? Currently it seems that you excluded from your analysis all the later dates except the oldest one?

Unfortunately, we do not believe it is possible to estimate the “rate of recovery” using the method suggested by your reviewer, for two reasons:

 The parameter of interest for our model is the loss of “rock art” related skills in a specific population, measured on an ecologically realistic timescale (tens or hundreds of years). However, populations are mobile. Thus disappearance of rock art at a site does not necessarily represent the loss of the relevant cultural traits from the population - the population may simply have moved. Conversely, the presence of rock art over a long period of time does not necessarily imply continued rock art production by a single population - it may reflect production by multiple populations occupying the site in sequence.

 Given the fragmentation of the archaeological record, it is difficult to determine if the disappearance of rock art after a certain date is due to the loss of relevant cultural traits or to accidents of ecology, preservation and excavation. 

In view of these considerations, we prefer to estimate the recovery rate using the methods described in our manuscript.

Final comments

The bottom line is that I still think it's worth publication, but that the limitations need to be discussed and highlighted early on. See my comments below and summary at the end.

Your reviewer is correct that the available rock-art data is sparse and that estimates of ancient population densities are necessarily crude. And, as he/she points out, the archaeological record is almost certainly biased. The revised version of our manuscript includes new analyses designed to give a clearer picture of these issues. However, we fully accept that no analysis can completely obviate the problems indicated by your reviewer. As mentioned earlier, we have revised the abstract, the discussion of bias and the conclusions , to recognize this reality. 

Reviewer 4

…. the authors ...tried to run the experiment with different relations between group interaction and population density (introducing a parameter $\\phi$ to generalise their model). They say the results are the same but also that they can't show the results. Would it be possible to have them at some point? recalculating the Table 1 for, let say, 3 different values of $\\phi$ and add that to the SI should not be that hard or I am wrong?

We agree with your reviewer that it is useful to show the results from this computation, which are now included in the last three columns of Table 1. We tested the model for phi=1.5, phi=2.5 and phi=6 (an extreme value). Choosing different values for phi slightly changes the most likely value for the critical threshold. However this is strongly constrained by the data and the changes are consequently small. Although the original model with phi=2 fits the data slightly better than the alternative models, the differences are insufficient to draw firm conclusions about the most likely value of phi. We have added text t (lines 284 - 289) illustrating these results.

Following on this interaction part of the epidemiological model, the authors also decide to "normalized" the parameter $\\beta$ to one. This sounds like a strong move to me, as if I understood correctly, this is the parameter representing the "virality" of new cultural variants, _ie_ the rate at which a cultural traits it's shared between populations. Thus, one would expect than with higher $\\beta$, the need for denser population will decreases. 

Your reviewer is correct that that beta is an indicator of virality or “infectivity” and that the higher its value the lower is the need for denser populations. The implication is that easily transmitted innovations (e.g., innovations that are relatively simple) can become endemic at lower population densities than those that are less infective (e.g. innovations that are more complex) 

Then, one could argue that by setting $\\beta$ this way the authors are forcing the model to work only with denser populations, while other path to CCE may be possible. ...the authors should spend a few more words to justify there choice here (maybe explaining why they think this won't change the overall outcome of the model if they think so? :

 The model testing procedure described in our paper determines a single value for the critical level of population density beyond which an innovation becomes endemic. 

Because the model predictions depend only on ρ^*, it is not possible to determine the individual values of α, β and γ but only the value of ρ^* . In an analysis where ρ^*is defined to take a single value, curve fitting will automatically compensate for changes in the value of βby changing the values of γand α. As a consequent normalization of the value of β to 1, has no impact on the value of the critical population density. 

On a side note, the value of beta is highly relevant for analyses where ρ^* can assume multiple values - e.g., analyses comparing transmission dynamics for different classes of cultural innovation

.

It seems that the Figure 1 wasn't included with the publication... 

We apologize for this omission. Figure 1 is now included.

I understood that Figure 3B (and some elements of 3A) shows the values for the "threshold", which is called $\\rho^{*}$ throughout the main text, but it's reported as $r^{*}$ in this figure. I may be wrong and $r^{*}$ is something else, but then I was not able to find what it was referring to. Also, I found the left panel of this figure confusing (the A panel) : 

Our revised manuscript includes a new version of Figure 3, where the threshold is correctly labeled. We have also rearranged Panel A, hopefully making it easier to read, and including the posterior distribution for the threshold.

 I have some questions regarding the model and the Bayesian inference. it would be nice to have an idea of those values, maybe a table summarising all parameters inferred, with a brief description of what they mean and their prior distribution 

As suggested by your reviewer, our revised manuscript now includes a Table (SI Table 2) showing the priors used for parameter estimation. The posterior distribution for $\\rhos^{*}$ is shown in Figure 3A. Unfortunately we were unable to devise a helpful representation of the joint posterior probabilities for the three parameters in our model.

It would help also to have a few sentences explaining how the likelihood is computed.

In response to this, and a similar request from reviewer 3, we have added a short paragraph describing the details of the computation (lines 489-502).

... minor problems popped up when I first tried to use the python code from the git repository (typos in folders' name and necessary packages that were not mentioned in the readme.txt) 

We have released a new version of the software where these problems are hopefully corrected. We have also corrected some of the variable names to align them with the terminology used in the paper. The new release allows users to reproduce all the new analyses included in the revised version of our manuscript. It is freely available, as of today, at https://github.com/rwalker1501/cultural-epidemiology.git. 

Minor typos

We have corrected our reference "Qitdlarssuaq: L’histoire d’une migration polaire". It appears that our reference manager cannot read French. Unfortunately we were not able to find the problem with the parentheses in line 150.

Other corrections

On rereading our manuscript and performing the new analyses requested by your reviewers we found two issues where the original version was unclear or inconsistent.

 The manuscript (and the associated software) were not always consistent in the nomenclature for cells. In the previous version of our manuscript we compare the frequency distribution of sites against the frequency distribution of “all cells”. In reality we were comparing it against the distribution of “non-sites”. In the new version of the manuscript we make this explicit. For practical purposes, the two distributions are indistinguishable. However, the new terminology is more accurate formally, and, we think, clearer for the reader.

 In our previous analysis we were not consistent in the labelling and definition of the geographical area previously referred to as “France-Spain”. We have now expanded the definition to formally include sites located in Portugal and repeated the analysis on this basis.

Hoping that you and your reviewers will appreciate this revised version of our manuscript

Yours sincerely

Richard Walker

(corresponding author)

---

## [Decision Letter · Decision Letter 2]

20 Jan 2021

PONE-D-19-31833R2

Stabilization of cultural innovations depends on population density: testing an epidemiological model of cultural evolution against a global dataset of rock art sites and climate-based estimates of ancient population densities.

PLOS ONE

Dear Dr. Walker,

Thank you for submitting your manuscript to PLOS ONE. As you can see below, both reviewers recommended publication of your ms. I do agree with them, but would like you to consider the following two suggestions before proceeding towards publication:

- Reviewer 4 inquired the authors about possible interpretations of the new results obtained from new $\\phi$ values. Including a sentence or two about the effect of the nature of the social interactions within the studied societies might be interesting.

- The current structure of the manuscript (with the Methods section at the end) imposes a brief introduction of the model and the dataset at the very beginning of the Results section. PLOS ONE is flexible regarding text structure, but locating the Methods section between the introduction and the Results section could help to avoid these redundancies.

We look forward to receiving your revised manuscript.

Kind regards,

Sergi Lozano

Academic Editor

PLOS ONE

Reviewers' comments:

Reviewer's Responses to Questions

**Comments to the Author**

1. If the authors have adequately addressed your comments raised in a previous round of review and you feel that this manuscript is now acceptable for publication, you may indicate that here to bypass the “Comments to the Author” section, enter your conflict of interest statement in the “Confidential to Editor” section, and submit your "Accept" recommendation.

Reviewer #3: All comments have been addressed

Reviewer #4: All comments have been addressed

2. Is the manuscript technically sound, and do the data support the conclusions?

Reviewer #3: (No Response)

Reviewer #4: Yes

3. Has the statistical analysis been performed appropriately and rigorously? 

Reviewer #3: (No Response)

Reviewer #4: Yes

4. Have the authors made all data underlying the findings in their manuscript fully available?

Reviewer #3: (No Response)

Reviewer #4: Yes

5. Is the manuscript presented in an intelligible fashion and written in standard English?

Reviewer #3: (No Response)

Reviewer #4: Yes

6. Review Comments to the Author

Reviewer #3: The authors have made an effort to address the possible caveats and concerns that I highlighted.

Although I am still somewhat skeptical about the ability to support or reject the proposed model based on such noisy and sparse data (e.g. it isn't discussed, but the error bars on the population sizes should be huge), I think the manuscript is sound enough at this point that it can be published and the reader should assess for herself whether she finds it sufficiently convincing.

The slight attenuation that the authors have done of the decisiveness in which the conclusions are stated, both in the abstract and later on, is appropriate.

Reviewer #4: The authors addressed all the comment made in the previous review. They added valuable explanation regarding the computation of the likelihood, as well as the computation of the AIC criterion which give even more credit to their findings. The authors also re-ran simulations with different $\\phi$, showing that it gives pretty similar results, but sadly don't interpret it ; I am curious to know if the authors have an idea of what this implying regarding the nature of the social interactions within the societies studied.

Aside this I think the manuscript is good to be published in PLOS ONE and will be a great addition to the (heated) debate regarding population size and cultural evolution.

7. PLOS authors have the option to publish the peer review history of their article (what does this mean?). If published, this will include your full peer review and any attached files.

Reviewer #3: No

Reviewer #4: No

---

## [Author Response · Author response to Decision Letter 2]

16 Feb 2021

In answer to the query from Reviewer 4, we have extended our discussion of the results we obtained by varying the value of $\\phi$ in our generalized model (see lines 262-286). On the basis of our results, we suggest that it is unlikely that high population mobility can completely compensate the negative effects of low population density, at least in the case of rock art. It is nonetheless likely that high mobility and long-distance social contacts can compensate for some of these effects, creating Culturally Effective Populations that are much larger than the census population of a specific area, as suggested in reference 19.

In reply to a useful suggestion from the editor, we have tried to eliminate a number of redundancies between the Introduction, the Methods and the Results. To this end, we have eliminated the description of the model from the Methods, moved the formal derivation of the model to Supporting Information, and introduced a concise description of the model immediately after the Introduction. As in the previous version, we have placed the Methods at the end of the article. We believe that this approach will make the article more accessible for non-specialist readers.

---

## [Editor Report · Decision Letter 3]

18 Feb 2021

Stabilization of cultural innovations depends on population density: testing an epidemiological model of cultural evolution against a global dataset of rock art sites and climate-based estimates of ancient population densities.

PONE-D-19-31833R3

Dear Dr. Walker,

We’re pleased to inform you that your manuscript has been judged scientifically suitable for publication and will be formally accepted for publication once it meets all outstanding technical requirements.

Kind regards,

Sergi Lozano

Academic Editor

PLOS ONE
---

## [Editor Report · Acceptance letter]

22 Feb 2021

PONE-D-19-31833R3 

Stabilization of cultural innovations depends on population density: testing an epidemiological model of cultural evolution against a global dataset of rock art sites and climate-based estimates of ancient population densities 

Dear Dr. Walker:

I'm pleased to inform you that your manuscript has been deemed suitable for publication in PLOS ONE. Congratulations! Your manuscript is now with our production department. 

Kind regards, 

on behalf of

Dr. Sergi Lozano 

Academic Editor

PLOS ONE